# Efficacy and Mechanism of Polymerized Anthocyanin from Grape-Skin Extract on High-Fat-Diet-Induced Nonalcoholic Fatty Liver Disease

**DOI:** 10.3390/nu11112586

**Published:** 2019-10-27

**Authors:** Meiqi Fan, Young-Jin Choi, Yujiao Tang, Sung Mun Bae, Hyun Pil Yang, Eun-Kyung Kim

**Affiliations:** 1Division of Food Bioscience, College of Biomedical and Health Sciences, Konkuk University, Chungju 27478, Korea; fanmeiqi@kku.ac.kr (M.F.); choijang11@kku.ac.kr (Y.-J.C.); yuanxi00@126.com; (Y.T.); 2Changchun University of Science and Technology, Changchun 130-600, China; 3Gyeongnam Agricultural Research and Extension Services, Jinju 52733, Korea; smbae@korea.kr; 4Technical R and D Center, Kitto Life Co., Ltd., Pyeongtacek 17749, Korea; yanghp0419@naver.com

**Keywords:** polymerized anthocyanin, nonalcoholic fatty liver disease, fat accumulation

## Abstract

We investigated the therapeutic potential of polymerized anthocyanin (PA) on a nonalcoholic fatty liver disease (NAFLD) model in mice. C57BL/6 mice were fed a high-fat diet (HFD) for 8 weeks to establish the NAFLD mouse model and randomly divided into four groups: control diet (con), NAFLD mice treated with saline (NAFLD), NAFLD mice treated with PA (PA), and NAFLD mice treated with orlistat (Orlistat) for four weeks. Mice were euthanized at the end of the four weeks. Total cholesterol (TC) and triglyceride (TG) levels were estimated, and pathological changes in the liver, white adipose tissue, and signaling pathways related to lipid metabolism were evaluated. Results revealed that the body, liver, and white fat weight of the NAFLD group was significantly increased compared to that of the con group, while that of the PA group showed significant reduction. NAFLD led to an increase in blood lipids in mice (except for HDL). Conversely, PA effectively reduced TC and LDL-C. Compared to the control group, the degree of steatosis in the mice of PA group was decreased. Moreover, PA also regulated the NAFLD signaling pathway. In agreement with improved lipid deposition, PA supplementation inhibited the activation of inflammatory pathways, depressing oxidative stress through increased antioxidant levels, and increasing β-oxidation to inhibit mitochondrial dysfunction. Taken together, our results demonstrate that PA can improve the liver function of NAFLD mice, regulating blood lipids, reducing liver-fat accumulation, and regulating lipid metabolism.

## 1. Introduction

Accumulating evidence indicates that excessive intake of fat is associated with the risk of developing nonalcoholic fatty liver disease (NAFLD) in humans and animals [1]. NAFLD is a stress-related liver injury and metabolic disorder characterized by diffuse hepatic steatosis and triglyceride accumulation [2,3], resulting in simple fatty liver, steatohepatitis, and cirrhosis [4]. The global prevalence of NAFLD is approximately 30% [5]. In patients with obesity and diabetes, the prevalence is as high as 70%–80%, which is a serious threat to public health [6]. NAFLD not only causes hepatic-cell steatosis and affects the liver’s metabolic function, it further progresses into cirrhosis and leads to metabolic diseases such as diabetes [7,8].

The occurrence and development of NAFLD can be related to several factors, such as lipid accumulation, inflammatory factors, abnormal expression of adipokines, gut microbiota, genetic predisposition, and oxidative stress (OS) [8,9]. To understand NAFLD pathogenesis, the existing "two-hit", "multiple-hit", and "four-step" doctrines are still not perfect, but the most classical "two-hit" doctrine explains the pathogenesis of the disease to an extent [10,11,12]. In the two-hit model, lipid accumulation in hepatocytes is the first hit and forms the basis for disease occurrence as it increases the vulnerability of the liver to several factors that might constitute the second hit [13].

Currently, the United States Food and Drug Administration (FDA) has not authorized any specific drugs for the treatment of NAFLD [14]. With the field moving rapidly towards combination therapies, single drugs with multiple cellular or molecular targets of action are under evaluation [15]. Orlistat is used as a weight-loss agent because it induces fat malabsorption by inhibiting enteric and pancreatic lipase [16]. A randomized controlled trial showed that orlistat improved hepatic steatosis in obese NAFLD patients [17]. Recently, there has been great interest in finding effective compounds from natural sources to treat NAFLD owing to their low toxicity and potentially fewer side effects [18,19].

Numerous studies have revealed the beneficial effects of wine on human health contributed by its antioxidant, anticarcinogenic, and cardiovascular-protection properties [20,21,22]. Grape skin is a waste byproduct of the wine industry, and studies have identified the potential of grape-skin extract in improving lipid metabolism in adipose tissue and liver in response to a high-fat diet (HFD) [23]. Studies have revealed that grape skin contains high levels of anthocyanin, a major polyphenol with antihyperglycemic and antioxidant properties [24]. Reports also indicate that anthocyanins from other sources may have protective effects against obesity-related NAFLD [25]. Polymerized derivatives of anthocyanins are more active than the monomeric forms. For example, polymerized anthocyanin (PA), derived from cranberry fruits, such as small anthocyanin glycosidic polymers, has higher antioxidant activity than the monomers, especially in the dimer, trimer, tetramer, and pentamer forms. These are highly water-soluble and fat-soluble compounds, and they do not accumulate in the body [26]. We have previously demonstrated successful PA synthesis from grape-skin extract by *Aspergillus niger* fermentation [27,28]. In another experiment, we examined the insulin resistance status and oral glucose tolerance test. The results showed the potential of PA in improving diabetes (Appendix A).

In the present study, we investigated the effect of PA on NAFLD induced by HFD in mice. We specifically evaluated the effects of PA on TG-associated lipogenic factors and cholesterol synthesis, as well as the regulatory factors of oxidative stress, inflammation, and mitochondrial dysfunction.

## 2. Materials and Methods 

### 2.1. Polymerized-Anthocyanin Synthesis

The synthesis method is shown in Figure 1A. *Aspergillus niger* can induce polymerization at the 4–8 bond position of the anthocyanin unit. The average molecular weights and the distribution of nonpolymerized and polymerized anthocyanin were measured by gel permeation chromatography (GPC, Tosoh, Germany) using a sodium nitrate (0.02 N, pH 7) as elution solvent. The samples were prepared with the following method: 3 mg of nonpolymerized and polymerized anthocyanin each was dissolved in 1 mL of sodium nitrate, and then filtered by 0.2 μm syringe filter. Then, 10 μL of the final sample was injected and measured to a condition of under 40 °C and a flow rate of 0.35 mL/min. As shown in Figure 1B,C, the molecular weight of the nonpolymerized anthocyanin was found to be 788 Da, whereas the molecular weight of the PA was 2255 Da. In addition, polydispersity (PDI), which shows homogeneous molecular-weight distribution, was 1.28. That means the PA contained homogeneous molecular-weight distribution, and the anthocyanin was consistently polymerized.

### 2.2. Experiment Animals, Diet, and Treatments 

Four-week-old male C57bl6/J mice were purchased from the Nara Bio creature focus (NARA Biotech, Seoul, Korea) and housed under specific pathogen-free (SPF) conditions. The mice were acclimatized for 1 week, housed in plastic cages, fed on standard chow diet with access to water ad libitum in a controlled environment, with a 12/12 h light/dark cycle, temperature of 20–21°C, and relative humidity of 40%–45%. All experiments were approved by the Institutional Animal Care and Use Committee (IACUC) of Konkuk University (KU19177), and every effort was made to limit the suffering of the animals and the number used in this investigation.

After acclimatization, the mice were randomly divided into 2 groups: control group (*n*  =  8) and NAFLD group (*n*  =  30). Mice in the control group received a standard diet (10% kcal fat), while mice in the NAFLD group received a high-fat diet (60% kcal fat). Food consumption was recorded every day, total energy intake was calculated according to the energy of different feeds after the animal experiment, and their body weight was evaluated twice every 7 days. After 8 weeks, mice with a body weight 20% higher than the initial weight were chosen and randomly divided into 3 groups (*n* = 8): NAFLD mice treated with saline, NAFLD mice treated with PA (NAFLD + PA; 400 mg/kg), and NAFLD mice treated with orlistat (NAFLD + Orlistat; 60 mg/kg) for approximately 1 month. Orlistat was used as a positive control. All samples that were dissolved in 100 μL of deionized water were given orally once a day for 4 weeks. After 12 weeks, the mice were subjected to medium-term fasting and sacrificed (Figure 1D).

Blood was collected for further investigation. Fat tissue (epididymal, subcutaneous, visceral, interscapular) was weighed, and epididymal adipose and liver were collected, frozen in liquid nitrogen, and stored at −80°C until further investigation. Samples of epididymal fat and liver were also fixed in 10% formaldehyde for further histological examination.

### 2.3. Body-Fat-Composition Analysis

Dual-energy X-ray Absorptiometry (DXA) was used to estimate muscle-to-fat ratio in the mice. DXA estimations were performed after 1 month under the diets described in the previous paragraph by using a complete body scanner (InAlyzer dual X-ray absorptiometry, Medikors, Gyeonggi, Republic of Korea). DXA measurement was performed once at low sensitivity and once at high sensitivity to discriminate between bones and tissue measured in gram units, the latter of which divided into fat and lean mass before examination as described by the manufacturer.

### 2.4. Histological Analysis

Liver and epididymal fat tissue were cut, fixed in 10% formalin, and embedded in paraffin for histological assessment. The formalin-fixed and paraffin-embedded tissue were cut into 4 µm thick slices and stained with hematoxylin and eosin (H and E). Random areas were observed under an optical microscope at 200× magnification. Liver cells were examined for degeneration (edema, steatosis) and necrosis. There were no hepatic sinus dilation and congestion, no fibrous tissue hyperplasia, and inflammatory cell infiltration in the portal area. Adipocyte sizes were assessed using a microscope. 

### 2.5. Blood Biochemical Analysis

Blood samples were obtained by cardiac puncture, and serum was separated by centrifugation (3000 rpm for 20 min) and stored at −80 °C until assayed. Serum levels of alanine aminotransferase (ALT), alkaline phosphatase (ALP), aspartate aminotransferase (AST), total protein (TP), serum albumin (ALB), and electrolytes were measured using an automatic analyzer (Abaxis VETSVAN VS2 Chemistry Analyzer, CA, USA). Total cholesterol (TC), low-density lipoprotein cholesterol (LDL-C), high-density lipoprotein cholesterol (HDL-C), and triglyceride (TG) were measured using a rapid blood lipid analyzer (OSANG healthcare Lipid Pro, Anyang, Korea). Leptin was quantified using a leptin ELISA kit (MERCK, DHB, GER).

### 2.6. mRNA Expression

Total RNA was isolated from the liver tissue of mice belonging to each treatment using TRIzol according to the manufacturer’s protocol. cDNA was obtained using Superscript II switch transcriptase (Invitrogen). RT-qPCR was performed on a Warm Cycler Shakers TP850 (Takarabio Inc., Shiga, Japan) according to the manufacturer’s protocol. RT-PCR conditions were the same as previously reported by Kim et al. [29]. Briefly, 2 μL cDNA (100 ng), 1 μL sense and antisense primers (0.4 µM), 12.5 μL SYBR Premix Ex Taq (Takarabio Inc.), and 9.5 μL of dH2O were mixed to obtain a 25 μL solution. PCR primers used for gene-expression analysis are listed in Table 1. Amplification conditions were as follows: 10 s at 95 °C, 40 cycles of 5 s at 95 °C, 30 s at 60 °C, 15 s at 95 °C, 30 s at 60 °C, and 15 s at 95 °C. GAPDH was used as the reference gene, and mRNA relative expression of the target genes was obtained by the formula: ΔCt= Ct of target gene−Ct of GAPDH.

### 2.7. Protein Expression

Protein expression was evaluated by western blot. Frozen mouse epididymal fat tissue was homogenized in liquid nitrogen. Tissue was lysed in RIPA lysis buffer (Sigma) containing 1% PI. Cell lysate was kept on ice for 1 h and subsequently centrifuged at 13,000 rpm for 20 min at 4°C. Total proteins (30 μg protein/test) were isolated by 10% sodium dodecyl sulfate-polyacrylamide gel electrophoresis (SDS-PAGE) and transferred to a nitrocellulose membrane (DC, Invitrogen). The membrane was blocked with 5% skim milk and subsequently incubated at 4°C with the following primary antibodies (1:5000 dilution): anti-AMPK, anti-SREBP-1c, anti-FAS, anti-ACC, anti-PPARγ, anti-C/EBPα, anti-FABP4, and anti-adiponectin (Cell Signaling Technology, Danvers, MA, USA). After washing, the membrane was incubated with IgG HRP-conjugated secondary antibody (1:2000) for 2 h at room temperature. Ponceau S was used for staining the protein bands. β-actin was used as the loading control. Proteins were visualized using enhanced-chemiluminescence (ECL) location reagent and quantified with the ImageJ program.

### 2.8. Statistical Analysis

Statistical analysis was performed using SPSS version 11.5 for Windows (SPSS Inc., Chicago, IL, USA). The mean of 2 continuous normally distributed variables was compared by independent sample Student’s test. Dunnett’s multiple-range tests were used to compare the means of 2 and 3 or more groups of variables not normally distributed. Values of *p* < 0.05 and *p* < 0.01 were considered statistically significant.

## 3. Results

### 3.1. Effects of PA on Body Composition in High-Fat-Diet-Induced NAFLD Mice

The impact of PA on body weight and fat composition was evaluated in mice under a high-fat-diet regimen. HFD increased adiposity, body weight, and caloric intake compared with the control diet that did not influence the weight, suggesting more pronounced digestion instead of changes in hunger (Table 2). Next, the effects of PA on HFD-induced body composition, fat mass, and body weight were evaluated by DXA. DXA results of the control, NAFLD, NAFLD with PA (400 mg/kg), and NAFLD with Orlistat (60 mg/kg) groups are displayed by the radiography of muscle-to-fat ratio (Figure 1E). The PA supplement for approximately 1 month essentially reduced the body weight in the HFD mice. The general dispersion of fat mass was also statistically reduced (Figure 1F,G; *p* < 0.05) during this treatment. Fat- mass arrangement in the NAFLD mice group appeared as increased liver size, white fat tissue (instinctive, subcutaneous, and epididymal), and interscapular darker fat tissue when contrasted with the control group. Interestingly, PA treatment fundamentally reduced white-fat-tissue weight and completely normalized brown-adipose-tissue weight (Table 2). 

### 3.2. Effects of PA on Lipid Accumulation and Hepatocellular Damage

To confirm the reduction in fat mass after PA treatment, histological investigation was performed on both epididymal fat tissue and liver. According to our results, PA had an hepatoprotective effect, and the outcomes from this investigation showed that lipid accumulation in the liver and fat tissue was essentially reduced in the NAFLD mice model (Figure 2A,B). In epididymal adipose tissue, PA supplementation reduced the size of adipocytes more prominently in the NAFLD compared with the control group (Figure 2C). Of note, the liver lobules and hepatocytes of CON mice were normal, and the hepatic cords were neatly arranged. Hepatocytes in the NAFLD group showed obvious fatty degeneration with unclear lobular boundaries and discorded hepatic cord structure. Furthermore, diffused vacuoles and balloonlike changes were observed in the liver tissue. Compared to the NAFLD group, lesions in the PA group and the Orlistat group were all reduced to different extents, the lesion area was significantly reduced, vacuoles and balloonlike changes were reduced, and lobular structures were restored to varying degrees (Figure 2B). Further, liver weight and body-weight ratios were significantly decreased and almost reached a normal level after PA treatment (Figure 2D,E).

### 3.3. Effects of PA on Blood Biochemical Parameters

Next, to verify the impact of PA on lipid parameters, biochemical parameters were evaluated. Our results showed that ALT, ALP, AST, TP, TC, LDL, and TG levels in the control-diet group were significantly lower than those of the NAFLD group (Table 2). The same parameters analyzed in the NAFLD with PA group showed significantly reduced plasma TP, TC, TG, and LDL-C levels (Table 2), while HDL-C levels were increased when compared to those of the NAFLD group. ALT, AST, and ALP were significantly reduced in the NAFLD group with PA treatment when compared to the normal-diet group (Table 2).

### 3.4. Effects of PA on Lipogenic Molecule Signal Pathway 

To better understand the molecular mechanism of An Oli on fat deposition, the mRNA and protein expression of the following signaling molecules were measured: adenosine 5‘-monophosphate (AMP)-activated protein kinase (AMPK), peroxisome proliferator-activated receptor γ (PPARγ), sterol regulatory element-binding protein 1c (SREBP-1c), diacylglycerol acyltransferase (DGAT), acyl-CoA synthetase (ACS), fatty acid synthase (FAS), acetyl-CoA carboxylase (ACC), 3-hydroxy-3-methylglutaryl-CoA reductase (HMGCR) and CCAAT/enhancer binding protein α (C/EBPα), and fatty acid binding protein 4 (FABP4). Adiponectin levels in the PA-fed mice were significantly higher (Figure 3A,F). PA reduced mRNA and protein levels of multiple lipid accumulation-associated transcripts. PA treatment led to reduction of DGAT, ACS, FAS, HMGCR, PPARγ, and C/EBPα mRNA levels (Figure 3A–F), whereas AMPK, SREBP-1c, FAS, ACC, PPARγ, FABP4, and C/EBPα protein levels were enhanced in NAFLD mice (Figure 3G–N).

### 3.5. Effects of PA on Inflammatory Molecules

To further delineate the mechanism of the polymerized anthocyanin in HFD-induced NAFLD, the effect of PA on mRNA expression of the following inflammatory factors was evaluated: sirtuin 1 (SIRT1), tumor necrosis factor-α (TNF-α), interleukin-1β (IL-1β), interleukin-6 (IL-6), interleukin-10 (IL-10), and monocyte chemoattractant protein-1 (MCP-1). TNF-α, IL-1β, IL-6, IL-10, and MCP-1 mRNA expression was significantly decreased in PA- and Orlistat-treated group as compared to their expression in the NAFLD group (Figure 4). SIRT1 expression was significantly increased in the PA and Orlistat group as compared to the NAFLD group. The concentration of leptin in the HFD with PA group was also significantly reduced compared to that of the HFD group (Figure 2H). 

### 3.6. Effects of PA on Oxidative Stress and Mitochondrial Dysfunction

Obesity and inflammation are closely linked, and the role of reactive oxygen species (ROS) is well-documented in this regard [30]. Mitochondrial and peroxisomal oxidation of unsaturated fats can cause the development of ROS. Furthermore, mitochondria additionally create free radicals in the respiratory chain that is combined with oxidative phosphorylation [31]. Therefore, the impact of PA on mRNA expression of several oxidative stress and mitochondrial dysfunction markers was evaluated, analyzing the following markers: nuclear factor erythroid-derived 2-related factor 2 (Nrf2), superoxide dismutase (SOD), catalase (CAT), glutathione peroxidase (GPx), glutathione reductase (GR), uncoupling protein 3 (UCP3), carnitine palmitoyltransferase I (CPT-1), and peroxisome proliferator-activated receptor-gamma coactivator 1-alpha (PGC-1α). Nrf2, SOD, CAT, GPx, GR, UCP3, CPT-1, and PGC-1α mRNA expression was significantly increased in the PA and Orlistat groups as compared to their expression in the NAFLD group (Figure 4B,C).

## 4. Discussion

Obesity, hyperlipidemia, and hypertension significantly increase the risk of NAFLD [32]. Despite active studies in the NAFLD field, its pathogenesis is still unclear. Currently recognized as the second-hit doctrine, fat accumulation is the primary factor in the development of hepatic steatosis. Excessive accumulation of fat in target organs and tissue such as the liver and adipose tissue causes abnormal lipid metabolism in hepatocytes and increased decomposition of peripheral fat, resulting in the first hit, the fatty degeneration of hepatocytes. Increased susceptibility to exogenous and endogenous damage, mitochondrial dysfunction, oxidative stress, and lipid-peroxidation damage further lead to the inflammation and necrosis of hepatocytes, resulting in the second hit [33]. 

### 4.1. Effects of PA on Body Composition and Pathomorphology

The establishment of an ideal animal model can accurately reflect the etiology, disease progression, and pathological features of human nonalcoholic steatohepatitis (NASH), and it is the basis for elucidating the pathogenesis and exploring treatment methods. The HFD mimics the modern Western diet structure, whose main energy intake comes from fat. In recent years, high-fat-fed nonalcoholic-fatty-liver animal models have been widely developed using simple methods with a high success rate, good stability, low mortality, and low cost [34,35]. In this experiment, long-term feeding of HFD induced obesity, hepatic steatosis, oxidative stress, and mitochondrial dysfunction, accompanied by the increased secretion of inflammatory factors, similar to the histopathological and pathophysiological characteristics seen in NAFLD patients.

Adipose tissue, especially visceral fat, is a key mediator of NAFLD. The importance of visceral fat in the pathogenesis of fatty liver has also been demonstrated in many animal models [36]. Recent evidence suggests that visceral adipose tissue is a metabolic and inflammatory device that regulates the metabolism of the brain, liver, muscles, and cardiovascular system [37]. Experiment results showed that the body weight and adipose-tissue weight of the NAFLD mice treated with PA were significantly reduced. Histopathology revealed that fat cells in the adipose tissue of the model group significantly increased; hepatic tissue showed obvious hepatic steatosis, the hepatic-lobule boundary was unclear, the hepatic-cord structure was disordered, and liver tissue showed diffuse mixed vacuoles and balloons. The observed change was the same as that seen in the human liver [38]. After four weeks of PA intervention, the area of liver-tissue damage was significantly reduced, vacuoles and balloonlike changes showed a decline, the accumulation of lipid droplets in the liver was reduced, and cell and lobular structures were restored to varying degrees. 

### 4.2. Effects of PA on Blood Biochemical Parameters

NAFLD is a metabolic-stress liver injury closely related to dyslipidemia. Epidemiological studies have shown that hyperlipidemia is an important risk factor for fatty liver, and 20%–92% of hyperlipidemia patients have fatty liver [39]. The two-hit theory is the theoretical basis of the NAFLD pathogenesis. The first attack causes fat to deposit in the liver. The decomposition of fat tissue releases free fatty acids that enter the liver tissue. At the same time, fat in food is absorbed by the intestine, and the blood lipids interact with/bind to synthetic lipoprotein, are dissolved, and enter the blood circulation [40]. Therefore, serum TP, TG, TC, HDL-C, LDL-C levels are closely related to fatty liver. When TG formation exceeds its output and the VLDL synthesis barrier at the same time, it affects TG transportation to the liver, and TG accumulates in the liver. This becomes the first hit in the disease, thus leading to the occurrence and development of the disease. HDL-C apolipoprotein is synthesized in the liver and participates in liver metabolic/detoxication processes. HDL is involved in reverse cholesterol transport of excess serum cholesterol to the liver, where it is converted to bile salts. Accumulation of LDL-C in cells is reduced by binding to vascular endothelial cells and smooth muscle cell receptors, ultimately reducing the deposition of lipids in the viscera [41,42]. This plays an important role in NAFLD prevention. Biochemical indicators for assessing liver damage caused by fatty liver are still lacking in specificity. Serum AST and ALT, the two most sensitive indicators of cell damage, are clinically used for the diagnosis of liver disease. ALT is a sensitive indicator of hepatocyte injury. During NAFLD, the structure and function of hepatocytes change due to excessive lipid deposition, and the fragility of hepatocytes increases with oxidative stress. Inflammatory reactions and destruction of the liver-cell membrane cause ALT to be released into the blood from the cytoplasm, leading to elevated serum levels [43]. At the same time, oxidative stress during lipid deposition can also cause damage to the mitochondrial membrane of hepatocytes, resulting in decreased membrane fluidity. AST is released into the blood cells from the mitochondria of liver cells, resulting in elevated serum AST levels. The first hit is mainly fat degeneration, and the inflammatory reaction progresses to the second-hit stage. Therefore, the increase of serum ALT and AST may indicate changes from functional damage to organic lesions, from initial hit to second-hit development. Experiment results indicated that the levels of serum TP, TG, TC, LDL-C, ALT, AST in the NAFLD group compared with the con group were significantly elevated. Decreased HDL-C levels were consistent with previous studies, suggesting disorders in lipid metabolism, damaged hepatocyte membranes and/or mitochondrial membranes, or necrotic hepatocytes in the NAFLD group. Compared to the NAFLD group, the levels of TP, TG, TC, LDL-C, ALT, and AST in the PA-treated group were all decreased to a certain extent, and the levels of HDL-C were significantly increased, suggesting that PA has a positive effect on lowering blood lipids, improving lipid metabolism, and protecting the plasma membrane of liver cells. 

### 4.3. Effects of PA on Adiponectin and Leptin

A series of regulatory peptides, such as leptin and adiponectin, secreted by fat cells are involved in various stages of liver damage during NAFLD pathogenesis. Adiponectin is an important factor in the regulation of glycolipid metabolism and homeostasis [44]. There is a significant correlation between the distribution and content of adiponectin and visceral fat. Adiponectin concentration decreases with increasing fat content [45]. Adiponectin can reduce free fatty acids (FFA) production by increasing mitochondrial β-oxidation, and inhibiting TG deposition and oxidative stress in the liver. Liver damage is reduced by inhibiting the expression of inflammatory factors. Leptin is an antiresistance hormone produced by white adipose tissue that prevents the intrusion and accumulation of free fatty acids into nonfatty tissue. The concentration of alizarin increases with the increase of adipose tissue and is proportional to the amount of body fat. The concentration of alizarin in the blood of obese people is usually higher, which may lead to leptin resistance, accumulation of intrahepatic fat, an increase in fatty acid concentration, and the synthesis of triglycerides. These promote fatty liver formation. Leptin is also one of the causative agents of inflammation and necrosis, participating in the "second hit" of NAFLD. Leptin was found to regulate the inflammatory response of fatty liver, promoting the decomposition and release of fat cells, leading to the differentiation of hepatic stellate cells, thereby accelerating NAFLD progression [46]. Experiment results showed that the level of alizarin was significantly increased, and the level of adiponectin was decreased in the NAFLD group compared with the con group. PA significantly increased adiponectin expression and decreased serum alizarin levels. This further confirms that PA has a preventive effect on nonalcoholic fatty liver.

### 4.4. Effects of PA on Lipid Accumulation

PPARγ is a nuclear receptor known as a major regulator in adipocyte biology [47]. Administrating NAFLD mice with a PPARγ agonist apparently reversed steatohepatitis [48]. Due to this reason, PPARγ was examined first in this study. Most genes in lipogenesis are bound by the induction of PPARγ and C/EBPα [48,49]. FABP4 is involved in adipocyte differentiation in the PPAR signaling pathway [50]. The function of FABP4 in fatty traits has been examined in many studies. It has been reported that TG content is also closely related to FABP4 activity [51]. TGs are the most important lipid type in the liver of patients with NAFLD and closely related to hepatic steatosis. Both ACS and DGAT are key factors in the TG synthesis pathway [52]. In the present study, our results demonstrated that PA downregulated the expression of mRNA of PPARγ, C/EBPα, ACS, and DGAT, and the protein levels of PPARγ, C/EBPα, and FABP4 were also significantly decreased. Adiponectin has been shown to activate almost all major types of target tissue, including skeletal muscle, liver, heart, vascular endothelial cells, adipocytes, and AMPK in the brain [53]. It has been reported that AMPK activation directly phosphorylates SREBP-1c in Ser372 residues, which results in the inhibition of SREBP-1c precursor cleavage and nuclear translocation, resulting I then inhibition of SREBP-1c-mediated lipogenesis [54,55]. SREBP-1c is mainly involved in fatty acid metabolism and glucose metabolism, and is a major transcriptional regulator of de novo lipogenesis (DNL). Glucose is converted to acetyl-CoA by glycolysis, and acetyl-CoA is converted to malonyl-CoA by ACC. Finally, under the action of FAS, acetyl-CoA and malonyl-CoA are catalyzed to form fatty acids, and the fatty acids can be esterified by ACS to synthesize triglycerides. In the case of excess energy, even if humans and animals do not consume fat, fat can be synthesized in large amounts by sugar through the de novo synthesis of lipids. The rate of lipogenesis is mainly regulated at the transcriptional level [56]. Cholesterol is mainly synthesized in the liver by a series of enzymatic reactions starting with acetyl-CoA as substrate. HMGCR is the rate-limiting enzyme in the synthesis reaction [57]. Our experiment results showed that PA can downregulate SREBP-1c. The RNA expression of its downstream target genes FAS and HMGCR, and the protein levels of SREBP-1, ACC, and FAS also showed a significant decrease. This indicates that PA improves the fatty acid-induced accumulation of lipid droplets in hepatocytes. At the same time, the cholesterol-lowering effect is achieved by inhibiting HMGCR.

### 4.5. Effects of PA on Inflammatory Molecules

The consequences of a single blow not only promote liver lipid deposition, but also increase the sensitivity of the liver to so-called secondary strokes such as inflammatory cytokines, mitochondrial dysfunction, oxidative stress, and endoplasmic reticulum (ER) stress [8]. The fatty liver is more susceptible to this “second hit”, and fibrosis can develop and ultimately lead to cirrhosis [11]. Inflammation are closely accompanied with lipid metabolism, are implicated in the pathogenesis of NAFLD, and further fibrosis and cirrhosis [58,59]. SIRT1, an NAD^+^-dependent protein deacetylase is an important regulator of energy homeostasis in response to nutrient availability [60]. When challenged with a high-fat diet, liver-specific SIRT1 knockout mice develop hepatic steatosis, hepatic inflammation, and ER stress [61]. In patients with NAFLD, the secretion of pro-inflammatory cytokines is increased (e.g., TNF-α, IL-6, IL-10, IL-1β, etc.). Specifically, the increase and activation of inflammatory cytokines such as TNF-α and IL-6 are important contents and key links of the "second strike" theory. TNF-α and IL-6 inhibit the expression of adiponectin and promotes the inflammatory response of nonalcoholic fatty liver [62]. The secretion of leptin exacerbates the inflammatory response and fibrosis of the liver during the progression of NAFLD [63]. IL-1β plays an important role in the development of NAFLD. Increased secretion of IL-1β promotes the cascade of proinflammatory cytokines TNF-α and MCP-1 [64] through a feedback loop. Activation of macrophages by fatty acids mediate hepatitis and liver damage. Increased secretion of chemokine MCP-1 can further promote hepatic macrophage infiltration [65]. Our results indicate that PA down-regulates the expression of mRNA for SIRT-1 and its downstream target genes TNF-α, IL-1β, IL-6, IL-10, and MCP-1. This indicates that PA plays a critical role in improving the inflammation associated with NAFLD.

### 4.6. Effects of PA on Oxidative Stress and Mitochondrial Dysfunction

Subsequently, the “second hit” emerges as oxidative stress and mitochondrial dysfunction, as well as inflammatory cytokines [11]. Oxidative stress refers to the rapid production of ROS in the body that exceeds the body’s ability to clear it and leads to tissue damage; this is one of the pathogenesis mechanisms of NAFLD [66]. In addition, the degree of oxidative stress is significantly associated with NAFLD severity, and the abnormal expression of oxidative-stress-related markers is often present in the liver of patients with NAFLD [67]. Nrf2 can regulate the expression of various antioxidant enzymes, including SOD, CAT, GPx, and GR. CAT and GPx synergize with SOD to form superoxide free radicals. SOD can avoid the adverse reactions caused by excessive ROS concentration in the body and is essential in the processing of ROS by reducing superoxide anions to form hydrogen peroxide. CAT and GPx further reduce hydrogen peroxide to water [68]. GPx is present in all tissue types of the human body, with its highest activity in liver tissue. GPx is an important component of the endogenous antioxidant system that can scavenge ROS and prevent oxidative-stress liver damage. A decrease in GR activity may be due to a decrease in GSSG levels because of decreased GPx activity. Loss of GR activity leads to a disruption of the GSH/GSSG ratio [69]. Our study found that the activities of Nrf2, SOD, CAT, GPx, and GR in the liver tissue of the NAFLD group were significantly decreased, indicating that there was oxidative damage in the liver tissue of NAFLD mice. By contrast, Nrf2, SOD, and CAT expression in the liver tissue of mice was higher in the PA group. GPx and GR activity was similarly significantly increased, suggesting that PA can effectively inhibit the oxidative-stress levels of liver tissue induced by a high-fat diet in the NAFLD mice. Experiment results indicated that PA downregulates the expression of mRNA for Nrf2 and its downstream target genes SOD, CAT, GPx, and GR. This indicates that PA plays a critical role in improving oxidative stress with NAFLD.

NAFLD can also inhibit the degradation of free fatty acids by directly inhibiting the β-oxidation process of free fatty acids [70], causing cytotoxic lipid substances (cholesterol etc.) to accumulate in the liver, leading to liver-cell and mitochondrial damage [71]. PPARγ induces the expression of mitochondrial proteins such as CPT-1 and UCP3. PGC-1α was initially identified as a nuclear PPARγ coactivator. Indeed, PPARγ can promote the expression of PGC-1α, which, in turn, potentiates PPARγ activity [72]. PGC-1 enhances fatty acid oxidation by promoting the production of CPT-1 [73,74]. Experiment results showed that PA downregulates the expression of mRNA for UCP3, CPT-1, and PGC-1α. This indicates that PA improves mitochondrial dysfunction in hepatocytes. This further confirms that PA has a preventive effect on nonalcoholic fatty liver. 

## 5. Conclusions

From the perspective of natural disease, the ideal drug for the treatment of NAFLD should be able to control the primary disease, reduce the accumulation of fat in the liver, prevent steatohepatitis, and delay the progression of fatty liver fibrosis and cirrhosis. However, to date, such a drug has not been discovered. In the present study, we conducted in-depth studies on the pathology, biochemistry, lipid accumulation, oxidative stress, inflammation, and mitochondrial gene expression of mouse models of fatty liver, and assessed the activity of PA as a potential drug by inhibiting the second hit following the first hit (Figure 5). However, it is difficult to identify a single major factor or mechanism responsible for the inhibitory effect of PA on hepatic lipid accumulation due to the complicated mechanism of lipid synthesis and metabolism. Therefore, further investigation is required.

## Figures and Tables

**Figure 1 nutrients-11-02586-f001:**
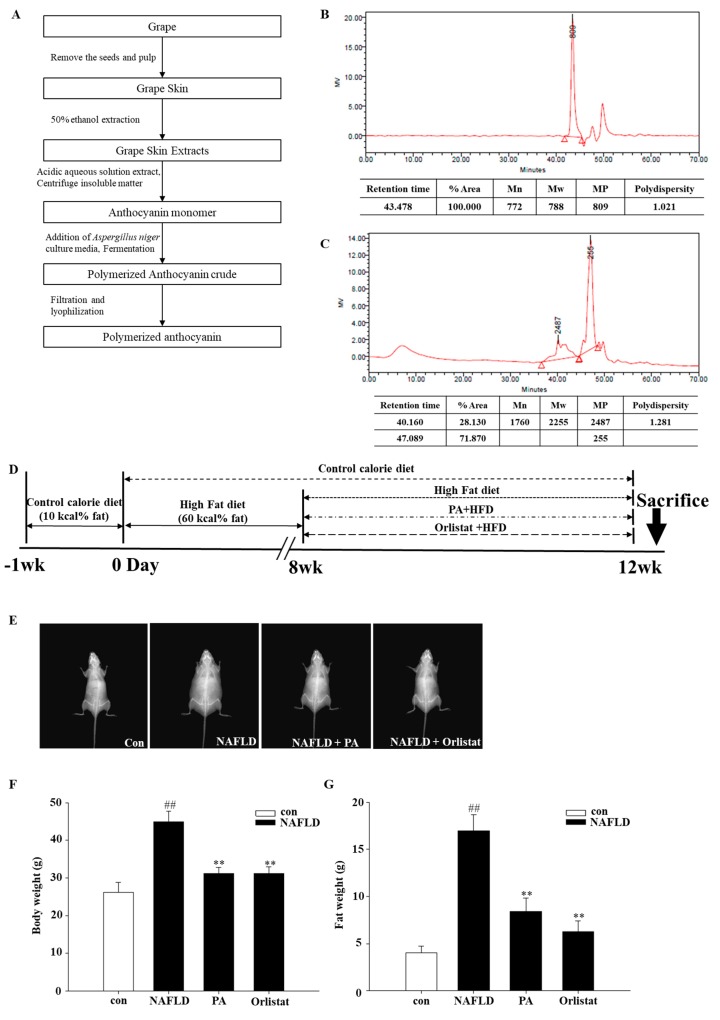
(**A**) Flowchart for the synthesis of polymerized anthocyanin using glucosidase from *Aspergillus niger*. Gel permeation chromatography (GPC) chromatogram of (**B**) nonpolymerized and (**C**) polymerized anthocyanin (PA). (**D**) Timeline for the in vivo study. Mice were randomly assigned into two groups and then fed either a control diet (con) or a high-fat diet for eight weeks. Mice with a body weight 20% higher than the con group were selected and then randomly divided into three groups: nonalcoholic fatty liver disease (NAFLD) mice treated with saline, NAFLD mice treated with PA (PA; 400 mg/kg), and NAFLD mice treated with orlistat (Orlistat; 60 mg/kg) for four weeks. Distribution of fat-weight and body-weight measurements by dual-energy X-ray Absorptiometry (DXA) on mice fed a control fat diet, high-fat diet, high-fat diet with 400 mg/kg PA, and high-fat diet with 60 mg/kg Orlistat. (**E**) Body-fat radiograph. (**F**) Body weight and (**G**) fat mass measured by DXA. Data are mean ± SEM. # *p* < 0.05, ## *p* < 0.05 compared with con group; * *p* < 0.05, ** *p* < 0.01 compared with NAFLD group.

**Figure 2 nutrients-11-02586-f002:**
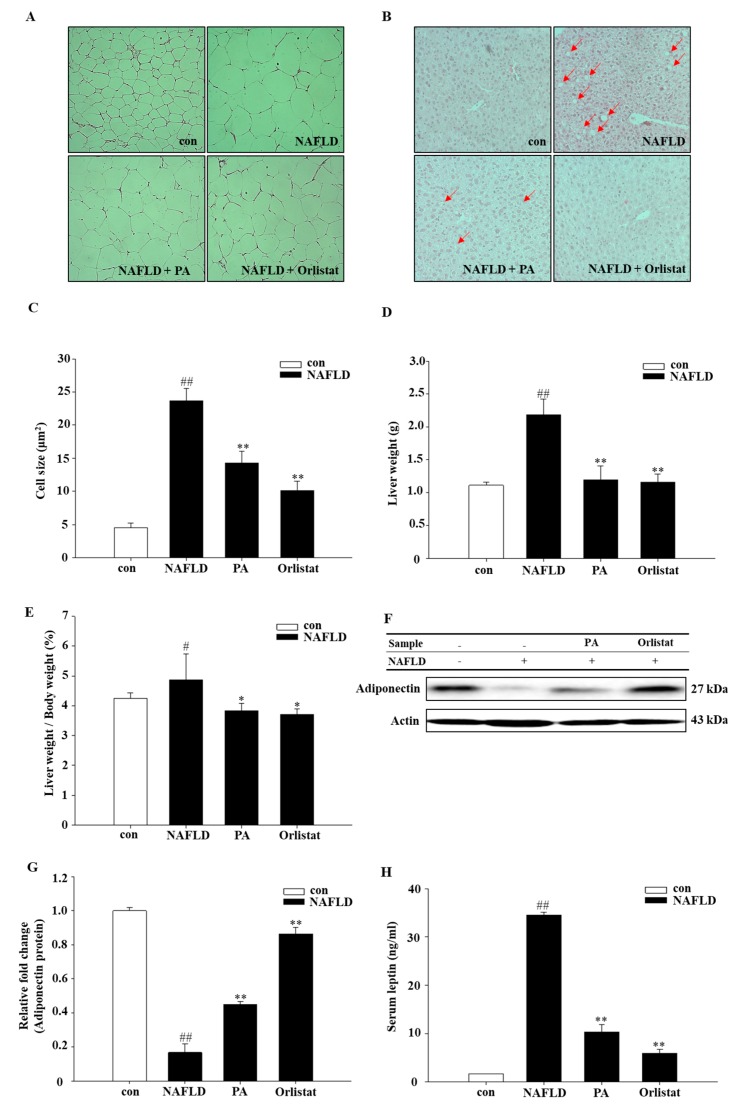
PA reduces lipid accumulation in liver and adipose tissue. Histological analysis of (**A**) epididymis adipose tissue and (**B**) liver based on hematoxylin and eosin staining. (**C**) Adipocyte mean area (μm^2^). (**D**) Liver weight (g). (**E**) Liver weight/body weight (%). Effect of PA on (**F**,**G**) adiponectin and (**H**) leptin. Data are representative of mean ± SEM of three independent measurements, # *p* < 0.05, ## *p* < 0.01 compared with the Con group; * *p* < 0.05, ** *p* < 0.01 compared with the NAFLD group.

**Figure 3 nutrients-11-02586-f003:**
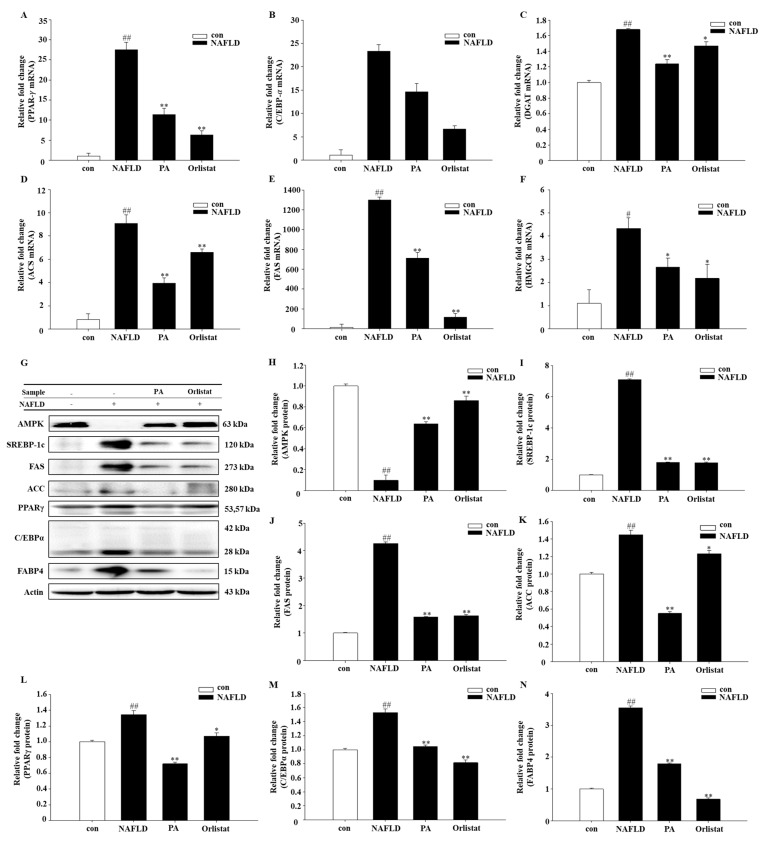
Effect of PA treatment on lipid accumulation in diet-induced NAFLD mice by real-time PCR. Graphs represent mRNA expression of transcription factors (**A**) peroxisome proliferator-activated receptor γ (PPAR-𝛾), (**B**) C/EBP-𝛼, (**C**) diacylglycerol acyltransferase (DGAT), (**D**) acyl-CoA synthetase (ACS), (**E**) fatty acid synthase (FAS), (**F**) 3-hydroxy-3-methylglutaryl-CoA reductase (HMGCR), (**G**) western-blot analysis, and AMPK (**H**), (**I**) sterol regulatory element-binding protein 1c (SREBP-1c), FAS (**J**), (**K**) acetyl-CoA carboxylase (ACC), (**L**) peroxisome proliferator-activated receptor gamma (PPARγ), (**M**) CCAAT/enhancer binding protein α (C/EBPα), and (**N**) fatty acid binding protein 4 (FABP4) protein expression levels. The relative protein expression data were normalized to that of actin. Data are representative of mean ± SEM of three independent measurements, # *p* < 0.05, ## *p* < 0.01 compared with the Con group. * *p* < 0.05, ** *p* < 0.01 compared with the NAFLD group.

**Figure 4 nutrients-11-02586-f004:**
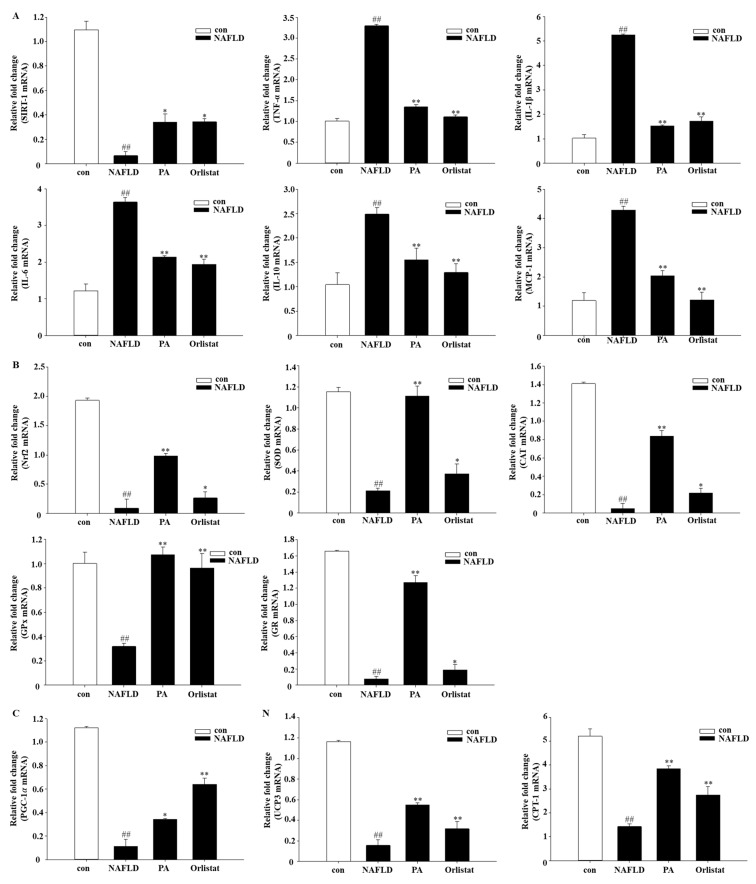
Effect of PA treatment on inflammatory molecules, oxidative stress, and mitochondrial dysfunction in diet-induced NAFLD mice by real-time PCR. Graphs represent mRNA expression of transcription factors: (**A**) Inflammatory molecules: sirtuin 1 (SIRT-1), tumor necrosis factor-α (TNF-α), interleukin-1β (IL-1β), interleukin-6 (IL-6), interleukin-10 (IL-10), monocyte chemoattractant protein-1 (MCP-1); (**B**) oxidative stress: nuclear factor erythroid-derived 2-related factor 2 (Nrf2), superoxide dismutase (SOD), catalase (CAT), glutathione peroxidase (GPx), glutathione reductase (GR); (**C**) mitochondrial dysfunction: peroxisome proliferator-activated receptor-gamma coactivator 1-alpha (PGC-1𝛼), uncoupling protein 3 (UCP3), carnitine palmitoyltransferase I (CPT-1). Data are mea n ±SEM. # *p* < 0.05, ## *p* < 0.01 compared with Con group; * *p* < 0.05, ** *p* < 0.01 compared with NAFLD group.

**Figure 5 nutrients-11-02586-f005:**
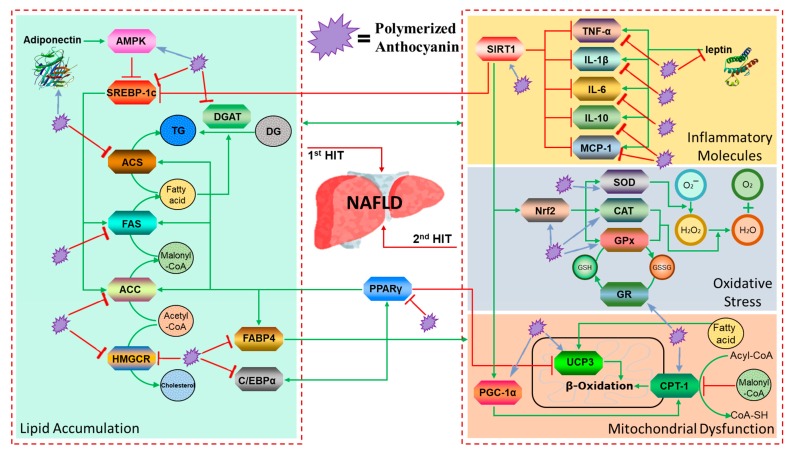
PA helps to prevent two-hit pathways in NAFLD through a variety of mechanisms, such as: (1) first hit: depressing lipid accumulation by downregulating lipogenesis factors; (2) second hit: ① inhibiting activation of inflammatory pathways, ② depressing oxidative stress through increased antioxidant levels, and ③ increasing β-oxidation to inhibit mitochondrial dysfunction.

**Table 1 nutrients-11-02586-t001:** Primers used in reverse transcriptase–polymerase chain-reaction analysis.

Gene Name		Sequence		Sequence
*PPAR-𝛾*	Forward	5-GAA AGA CAA CGG ACA AAT CAC-3	Reverse	5-GAA ACT GGC ACC CTT GAA-3
*C/EBP-𝛼*	Forward	5-CGT CTA AGA TGA GGG AGT C-3	Reverse	5-GGC ACA AGG TTA CTT CCT-3
*SREBP−1c*	Forward	5-CTT CTG GAG ACA TCG CAA AC-3	Reverse	5-GGT AGA CAA CAG CCG CAT C-3
*ACS*	Forward	5-AAG CCC AGA GTT ACG AGT AT-3	Reverse	5-ACA CAG GAA TAG AGG AGT TCT-3
*FAS*	Forward	5-CTT GGG TGC TGA CTA CAA CC-3	Reverse	5-GCC CTC CCG TAC ACT CAC TC-3
*HMGCR*	Forward	5-TTC TGC TCT TGA TTG ACC TTT C-3	Reverse	5-TTT CCC TTA CTT CAT CCT GTG A-3
*SIRT1*	Forward	5-GGC GGG GAA CGA CTG CG-3	Reverse	5-GGA GTC ATG GGG GCT GTA CTG-3
*TNF-α*	Forward	5-AAG CCT GTA GCC CAC GTC GT-3	Reverse	5-GGC ACC ACT AGT TGG TTG TC-3
*IL-1β*	Forward	5-AAC CAA GCA ACG AVA AAA TA-3	Reverse	5-AGG TGC TGA TGT ACC AGT TG-3
*IL-6*	Forward	5-CCG GAG AGG AGA CTT CAC AG-3	Reverse	5-GGA AAT TGG GGT AGG AAG GA-3
*IL-10*	Forward	5-TCA GCT GTG TCT GGG CCA CT-3	Reverse	5-TTA TGA GTA GGG ACA GGA AG-3
*MCP-1*	Forward	5-TGA TCC CAA TGA GTA GGC TGG AG-3	Reverse	5-ATG TCT GGA CCC ATT CCT TCT TG-3
*PGC-1α*	Forward	5-ATT CGG GAG CTG GAT GGC TT-3	Reverse	5-CCG ATT GGT CGC TAC ACC AC-3
*UCP3*	Forward	5-ACC CGA TAC ATG AAC GCT CC-3	Reverse	5-TCA TCA CGT TCC AAG CTC CC-3
*CPT-1*	Forward	5-TGT GTG AGG ATG CTG CTT CC-3	Reverse	5-CTC GGA GAG CTA AGC TTG TC-3
*Nrf2*	Forward	5-AGC ACA TCC AGA CAG ACA CCA GT-3	Reverse	5-TTC AGC GTG GCT GGG GAT AT-3
*SOD*	Forward	5-CAA TGG TGG GGG ACA TAT TA-3	Reverse	5-TTG ATA GCC TCC AGC AAC TC-3
*CAT*	Forward	5-GAA CGA GGA GGA GAG GAA AC-3	Reverse	5-TGA AAT TCT TGA CCG CTT TC-3
*GPx*	Forward	5-ACA TTC CCA GTC ATT CTA CC-3	Reverse	5-TTC AAG CAG GCA GAT ACG-3
*GR*	Forward	5-CGG CGA TCT CCA CAG CAA TG-3	Reverse	5-ACC GCT CCA CAC ATC CTG ATT G-3
*GAPDH*	Forward	5-GCA CAG TCA AGG CCG AGA AT-3	Reverse	5-GCC TTC TCC ATG GTG GTG AA-3

*PPAR-**𝛾*, peroxisome proliferator-activated receptor γ; *C/EBP-**𝛼*, CCAAT/enhancer binding protein α; *SREBP−1c*, sterol regulatory element-binding protein 1c; *ACS*, acyl-CoA synthetase, *FAS*, fatty acid synthase; *HMGCR*, 3-hydroxy-3-methylglutaryl-CoA reductase; *SIRT1*, sirtuin 1; *TNF-α*, tumor necrosis factor-α; *Il-1β*, interleukin-1β; *IL-6*, interleukin-6; *IL-10*, interleukin-10; *MCP-1*, monocyte chemoattractant protein-1; *PGC-1α*, peroxisome proliferator-activated receptor-gamma coactivator 1-alpha; *UCP3*, uncoupling protein 3; *CPT-1*, carnitine palmitoyltransferase I; *Nrf2*, nuclear factor erythroid-derived 2-related factor 2; *SOD*, superoxide dismutase; *CAT*, catalase; *GPx*, glutathione peroxidase; *GR*, glutathione reductase; *GAPDH*, glyceraldehyde-3-phosphate dehydrogenase.

**Table 2 nutrients-11-02586-t002:** Body, liver, and adipose-tissue weight.

	CON	HFD	HFD + PA	HFD + Orlistat
Initial body weight (g)	16.85 ± 0.58 ^a^	17.29 ± 0.44 ^a^	17.32 ± 0.72 ^a^	16.68 ± 0.93 ^a^
Final body weight (g)	26.14 ± 2.67 ^c^	44.86 ± 2.8 ^a^	31.27 ± 1.53 ^b^	31.20 ± 1.77 ^b^
Body-weight gain (g)	9.29 ± 2.43 ^d^	27.57 ± 2.55 ^a^	13.95 ± 3.6 ^c^	14.52 ± 3.21 ^b,c^
Total energy intake (kcal)	749.73 ± 26.02 ^c^	1194.97 ± 35.89 ^a^	973.68 ± 29.43 ^b^	975.14 ± 32.32 ^b^
Lean mass (g)	0.25 ± 0.07 ^d^	2.07 ± 0.64 ^a^	0.88 ± 0.39 ^b^	0.57 ± 0.24 ^c^
Epididymal-adipose-tissue weight (g)	0.20 ± 0.02 ^d^	3.50 ± 0.4 ^a^	1.02 ± 0.2 ^b^	0.75 ± 0.09 ^c^
Subcutaneous-adipose-tissue weight (g)	0.07 ± 0.05 ^d^	1.50 ± 0.08 ^a^	0.47 ± 0.04 ^b^	0.23 ± 0.14 ^c^
Visceral-adipose-tissue weight (g)	0.16 ± 0.03 ^c^	0.33 ± 0.08 ^a^	0.23 ± 0.02 ^b^	0.21 ± 0.04 ^b^
interscapular adipose tissue weight (g)	16.85 ± 0.58 ^a^	17.29 ± 0.44 ^a^	17.32 ± 0.72 ^a^	16.68 ± 0.93 ^a^
ALT (U/L)	34.50 ± 3.42 ^b^	66.00 ± 3.54 ^a^	39.00 ± 1.73 ^b^	35.00 ± 2.65 ^b^
AST (U/L)	73.50 ± 5.45 ^c^	344.50 ± 0.71 ^a^	125.75 ± 1.49 ^b^	129.33 ± 4.24 ^b^
TP (g/dL)	4.68 ± 0.59 ^c^	5.45 ± 0.64 ^a^	5.05 ± 0.38 ^b^	5.20 ± 0.50 ^a,b^
TC (mg/dL)	107.20 ± 4.21 ^d^	184.50 ± 3.27 ^a^	116.25 ± 5.29 ^c^	128.00 ± 4.97 ^b^
LDL-C (mg/dL)	21.40 ± 2.70 ^c^	114.00 ± 4.05 ^a^	45.50 ± 1.78 ^b^	46.00 ± 5.66 ^b^
HDL-C (mg/dL)	67.67 ± 4.02 ^a^	33.40 ± 2.70 ^c^	56.50 ± 3.54 ^b^	54.25 ± 2.35 ^b^
TG (mg/dL)	115.20 ± 3.0 ^c^	261.83 ± 3.32 ^a^	137.50 ± 2.36 ^b^	113.25 ± 1.66 ^c^

All data represent mean values ± SEM. ^a-h^ Values with different superscripts were significantly different with *p* < 0.05 as analyzed by Dunnertt’s multiple-range tests. ALT, alanine aminotransferase; AST, aspartate aminotransferase; TP, total protein; TC, total cholesterol; LDL-C, low-density lipoprotein cholesterol; HDL-C, high-density lipoprotein cholesterol, TG, triglyceride.

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
