# Peer review of "Efficacy and Mechanism of Polymerized Anthocyanin from Grape-Skin Extract on High-Fat-Diet-Induced Nonalcoholic Fatty Liver Disease"

_nutrients, 2019, doi:10.3390/nu11112586_

Round 1

Reviewer 1 Report

Hi,

There is a lot of data in your manuscript, and thank you for your hard work.  I have two main questions about your study.

1) Dietary oligomers or polymers are commonly known to act like dietary fiber to pull fat out of gut and reduce the impact of high fat diet.  Were fecal samples collected for lipid analysis?  It could be An-Oli pulled fat out of gut and signaling changed due to that, rather than An-Oli got into circulation and triggered systemic signaling changes.  Did your team look at that?

2) mRNA and protein expression levels vary from organ to organ, and liver, adipose and muscle are all impactful organs to lipid homeostasis.  For a liver disease research, why do you think the mRNA expression in mouse EAR tissue (not liver) is representative to explain inflammation / lipid gene changes across organs, and why protein expression in adipose is representative to explain fatty acid signaling changes across organs?

Below are minor suggestions

3) In your abstract, it will be good to summarize RNA/Protein data, which is a big chunk of data presented in this manuscript.  Animal model is meant to be a research tool to study mechanism and not efficacy.  While the observations of common biomarkers may be exciting, it is important to cover all major findings in abstract so readers can have the right expectation about what kind of data is in this manuscript.

4) How An-Oli and Orlistat were dose to mice?  It is not clear in the manuscript.  Is it oral gavage?  in what form and in what volume at what frequency?

5) It is not clear how DXA imaging software was set up to calculate lean mass(g) and various adipose tissue weight(g).  It will be good to provide some details in the method section otherwise this DXA machine becomes a black box  and not really replicable by others.

6) It is not clear how total energy intake data is collected.  Would you please describe in the material and methods section?

7) In table 2, ALT/AST/TP data were duplicated.

8) Discussions and figure 5 painted a picture of An-Oli induced systemic signaling changes related to non alcoholic fatty liver disease while the actual materials came from tissues other than liver.  This could be misleading...

Author Response

There is a lot of data in your manuscript and thank you for your hard work. I have two main questions about your study.

Thank you for your valuable time to review our manuscript. We are very much honored it and learned a lot from your supportive comments and useful suggestions. We also thoroughly revised whole manuscript as your comments, and our point-by-point reply to your comments was described as follows. In addition, other minor errors that we found while revising the manuscript were also corrected. Regarding the name of anthocyanin, according to our repeated thinking, we decided that, compared to the ‘Anthocyanin Oligomer’, ‘Polymerized Anthocyanin’ was more appropriate for the sample name (Please see Fig. 1). We have revised it in this paper and hope to get your understanding. For your convenience, we will use the name An-Oli in this response.

Point 1: Dietary oligomers or polymers are commonly known to act like dietary fiber to pull fat out of gut and reduce the impact of high fat diet. Were fecal samples collected for lipid analysis? It could be An-Oli pulled fat out of gut and signaling changed due to that, rather than An-Oli got into circulation and triggered systemic signaling changes. Did your team look at that?

Response 1: Thank you very much for your considerable comment. We agree with you. Dietary oligomers or polymers are commonly known to act like dietary fiber(DF) to pull fat out of gut and reduce the impact of high fat diet. This kind of dietary oligomers or polymers are usually can be defined as any nondigestible carbohydrate and lignin not degraded in the upper gut [1]. Major sources of DF are whole-grain cereals, fruit, vegetables, and legumes, which typically contain diverse types of DF. Whole-grain foods, by weight, generally contain some 12% of total (mainly insoluble cereal) DF, and there is a strong correlation between cereal DF and whole-grain consumption. Some bran-derived food products contain ≤25% DF [2]. The An-Oli used in this experiment was detected to be polyphenolic compounds, which would be absorbed through the intestinal tract. However, the impact of An-Oli on intestinal microorganisms is indeed worth investigating. We will explore this in the future research. We would appreciate it if you could understand. Once again, thank you for your valuable comments. Thanks to you, our investigation would be upgraded.

Weickert, M. O., & Pfeiffer, A. F. (2018). Impact of dietary fiber consumption on insulin resistance and the prevention of type 2 diabetes. The Journal of nutrition148(1), 7-12. Weickert, M. O., & Pfeiffer, A. F. (2008). Metabolic effects of dietary fiber consumption and prevention of diabetes. The Journal of nutrition138(3), 439-442.

Point 2: mRNA and protein expression levels vary from organ to organ, and liver, adipose and muscle are all impactful organs to lipid homeostasis. For a liver disease research, why do you think the mRNA expression in mouse EAR tissue (not liver) is representative to explain inflammation / lipid gene changes across organs, and why protein expression in adipose is representative to explain fatty acid signaling changes across organs?

Response 2: Thank you very much for your considerable comment. Thank you very much for your considerable comment. We are so sorry, that was our typo. We got the mRNA from the mouse liver tissue.

Lipid accumulation in the NAFLD hepatocytes and insulin resistance increase the vulnerability of the liver to many factors that act in a coordinated and cooperative manner to promote hepatic injury, inflammation and fibrosis. Among these factors, adipose tissue dysfunction plays a crucial role [1]. It has been shown that adipose tissue biology is much more complex than previously considered [2] and adipose tissue dysfunction has been proposed as a major contributor to NAFLD [3,4]. We tried to prove that an-oli functions the same in liver tissue and adipose tissue, so we conducted experiments on liver tissue and adipose tissue. Unfortunately, because of the special composition of adipose tissue, it is difficult to extract mRNA from adipose tissue. Therefore, we did a protein experiment to confirm that. Experiments have proved that the functions of an-oli in liver tissue and adipose tissue are consistent.

Cimini, F. A., Barchetta, I., Carotti, S., Bertoccini, L., Baroni, M. G., Vespasiani-Gentilucci, U., ... & Morini, S. (2017). Relationship between adipose tissue dysfunction, vitamin D deficiency and the pathogenesis of non-alcoholic fatty liver disease. World journal of gastroenterology23(19), 3407. Rosen, E. D., & Spiegelman, B. M. (2014). What we talk about when we talk about fat. Cell156(1-2), 20-44. Park, B. J., Kim, Y. J., Kim, D. H., Kim, W., Jung, Y. J., Yoon, J. H., ... & Jang, J. J. (2008). Visceral adipose tissue area is an independent risk factor for hepatic steatosis. Journal of gastroenterology and hepatology23(6), 900-907. van der Poorten, D., Milner, K. L., Hui, J., Hodge, A., Trenell, M. I., Kench, J. G., ... & George, J. (2008). Visceral fat: a key mediator of steatohepatitis in metabolic liver disease. Hepatology48(2), 449-457.

Below are minor suggestions

Point 3: In your abstract, it will be good to summarize RNA/Protein data, which is a big chunk of data presented in this manuscript. Animal model is meant to be a research tool to study mechanism and not efficacy. While the observations of common biomarkers may be exciting, it is important to cover all major findings in abstract so readers can have the right expectation about what kind of data is in this manuscript.

Response 3: Thank you very much for your considerable comment. We added the summary in the abstract section as your comment.

Point 4: How An-Oli and Orlistat were dose to mice? It is not clear in the manuscript. Is it oral gavage? in what form and in what volume at what frequency?

Response 4: Thank you very much for your considerable comment. We described in details about that as your comments.

Point 5: It is not clear how DXA imaging software was set up to calculate lean mass(g) and various adipose tissue weight(g). It will be good to provide some details in the method section otherwise this DXA machine becomes a black box and not really replicable by others.

Response 5: Thank you very much for your considerable comment. We revised the section as your comment. There are lots of DXA instruments in the world, and the software is different with company. We used ‘InAlyzer’ developed by Medikors Inc.. ‘InAlyzer’ can analyze Bone Mineral Density and Body Composition (BMC, FAT, LEAN) of alive lab animals in units of 0.001g with simple anesthesia without dissection. Serial assessment is possible for each entity of alive lab animals. We provided detailed instrument information so that others could replicable the experiment.

Point 6: It is not clear how total energy intake data is collected. Would you please describe in the material and methods section?

Response 6: Thank you very much for your considerable comment. Food consumption was recorded every day and the total energy intake was calculated according to the energy of different feeds after the animal experiment. We described in details about that in the material and methods section as your comment.

Point 7: In table 2, ALT/AST/TP data were duplicated.

Response 7: Thank you very much for your considerable comment. We corrected the table 2 as your comment.

Point 8: Discussions and figure 5 painted a picture of An-Oli induced systemic signaling changes related to non alcoholic fatty liver disease while the actual materials came from tissues other than liver. This could be misleading...

Response 8: Thank you very much for your considerable comment. The ‘EAR’ was our mistake. The actual materials came from liver tissue. Once again, appreciate your valuable comment.

Reviewer 2 Report

The authors in this paper have elucidated a potential protective role of  grape skin extract on HFD induced steatosis. The study is well designed and the authors have done great job supporting their conclusions. Few more editions in the manuscript will greatly improve the conclusions made by authors.

Did authors see similar protective effect with different diet models for NAFLD -NASH. eg. HPC diet or MCD diet ? Considering the HFD model induce fatty liver also depends on insulin resistance, the authors seem to have missed examining the insulin resistance status during the treatment.  The authors need to clarify at least in the discussion how the potential mechanism of grape extract mediated down regulation in different genes. Performing the experiments mentioned in point 1 and 2 will help understand the exact role. Based on the outcome the authors need to modify the discussion. along with liver weight I would like to see liver weight to body weight ratios. Preferably in the same figure. In figure 2 along with HE a oil red O staining will help strengthen the claims. Considering the effect on adiponectin. Authors should include genes involved in retinoic acid signaling pathway. Along with inflammatory gene a F4/80 IHC would be great ! this will help us understand the source of inflammation and exact effect. Presentation of  Figure 4 could be improved by grouping all inflammatory gene in one panel and oxidative in same panel.

Author Response

The authors in this paper have elucidated a potential protective role of grape skin extract on HFD induced steatosis. The study is well designed, and the authors have done great job supporting their conclusions. Few more editions in the manuscript will greatly improve the conclusions made by authors.

Thank you for your valuable time to review our manuscript. We are very much honored it and learned a lot from your supportive comments and useful suggestions. We also thoroughly revised whole manuscript as your comments, and our point-by-point reply to your comments was described as follows. In addition, other minor errors that we found while revising the manuscript were also corrected. Regarding the name of anthocyanin, according to our repeated thinking, we decided that, compared to the ‘Anthocyanin Oligomer’, ‘Polymerized Anthocyanin’ was more appropriate for the sample name (Please see Fig. 1). We have revised it in this paper and hope to get your understanding. For your convenience, we will use the name An-Oli in this response.

Point 1: Did authors see similar protective effect with different diet models for NAFLD -NASH. eg. HPC diet or MCD diet?

Response 1: Thank you very much for your considerable comment. As your comment, MCD diet, choline-deficient l-amino acid-defined (CDAA) diet, atherogenic diet, and high-fat diet [1] are the best described dietary models for NAFLD.

Although the MCD diet results in a rapid onset of the NASH phenotype with lobular inflammation and ballooning (2–8 weeks), the animals do not exhibit any other metabolic features that are seen in human NAFLD, including obesity, peripheral insulin resistance and dyslipidemia. On the contrary, animals fed an MCD diet show significant weight loss (up to 40% in 10 weeks) [2,3,4]. Therefore, this model is generally considered adequate to study the intrahepatic events in relation to NASH and the pharmacological treatment of NASH but is regarded as inadequate to study the multisystemic disease entity, that is NALFD, in all its aspects [5]. However, in the CDAA diet proteins are substituted with an equivalent and corresponding mixture of l-amino acids [6]. Animals fed a CDAA diet develop the same or perhaps a slightly more severe degree of NASH, as well as a larger increase in ALT levels, albeit on a marginally longer time frame. After 20–22 weeks, a significant amount of fibrosis is observed [7,8]. Although they do not experience the weight loss observed with the MCD diet, the metabolic features of human NAFLD still fail to appear when used in the same time frame as the MCD diet [2,9]. The Ath diet by itself does not induce weight gain, nor significant insulin resistance. Furthermore, epididymal fat pads, which are generally used as an experimental substitute for human visceral adipose tissue in rodents, appear to be smaller compared to animals fed standard chow (SC). As visceral adipose tissue plays an unmistakable role in NASH and other obesity-related conditions, including cardiovascular disease and DM2, this should certainly be considered a shortcoming [10].

The high-fat diet brings about a phenotype similar to the human disease, characterized by obesity (after 10 weeks), insulin resistance (hyperinsulinemia after 10 weeks and glucose intolerance after 12 weeks) and hyperlipidemia (after 10 weeks) [11]. The excess supply of free fatty acids, directly via intake and via increased lipolysis, brings about triglyceride accumulation in the liver [26]. Interestingly, steatosis develops after 1–2 weeks, but diminishes subsequently, only to reappear after 6–12 weeks [3,12]. Although NASH usually develops after 12 weeks, the observed steatosis and inflammation are substantially less pronounced than is the case in MCD diet-fed animals [4].

Among the above diet, due to the disease phenotype of NAFLD in humans, we selected the high-fat diet as the diet model and set the experimental period as 12 weeks. Hopefully, you satisfy our decision.

Van Herck, M., Vonghia, L., and Francque, S. Animal models of nonalcoholic fatty liver disease—a starter’s guide. Nutrients, 2017, 9, 1072. Ibrahim, S.H.; Hirsova, P.; Malhi, H.; Gores, G.J. Animal models of nonalcoholic steatohepatitis: Eat, delete, and inflame. Dig. Dis. Sci. 2016, 61, 1325–1336. Itagaki H., Shimizu K., Morikawa S., Ogawa K., Ezaki T. Morphological and functional characterization of nonalcoholic fatty liver disease induced by a methionine-choline-deficient diet in c57bl/6 mice. Int. J. Clin. Exp. Pathol. 2013, 6,2683–2696. Sanches S.C.L., Ramalho L.N.Z., Augusto M.J., da Silva D.M., Ramalho F.S. Nonalcoholic steatohepatitis: A search for factual animal models. BioMed Res. Int. 2015, 2015, 1–13. Byrne C.D., Targher G. Nafld: A multisystem disease. J. Hepatol. 2015, 62, S47–S64. Nakae D., Mizumoto Y., Andoh N., Tamura K., Horiguchi K., Endoh T., Kobayashi E., Tsujiuchi T., Denda A., Lombardi B., et al. Comparative changes in the liver of female fischer-344 rats after short-term feeding of a semipurified or a semisynthetic l-amino acid-defined choline-deficient diet. Toxicol. Pathol. 1995, 23, 583–590.  Kodama Y., Kisseleva T., Iwaisako K., Miura K., Taura K., De Minicis S., Osterreicher C.H., Schnabl B., Seki E., Brenner D.A. C-jun n-terminal kinase-1 from hematopoietic cells mediates progression from hepatic steatosis to steatohepatitis and fibrosis in mice. Gastroenterology. 2009, 137, 1467–1477.  Miura K., Kodama Y., Inokuchi S., Schnabl B., Aoyama T., Ohnishi H., Olefsky J.M., Brenner D.A., Seki E. Toll-like receptor 9 promotes steatohepatitis by induction of interleukin-1beta in mice. Gastroenterology. 2010, 139, 323–334. Ishioka M., Miura K., Minami S., Shimura Y., Ohnishi H. Altered gut microbiota composition and immune response in experimental steatohepatitis mouse models. Dig. Dis. Sci. 2017, 62, 396–406.  Matsuzawa N., Takamura T., Kurita S., Misu H., Ota T., Ando H., Yokoyama M., Honda M., Zen Y., Nakanuma Y., et al. Lipid-induced oxidative stress causes steatohepatitis in mice fed an atherogenic diet. Hepatology. 2007, 46, 1392–1403.  Cote I., Ngo Sock E.T., Levy E., Lavoie J.M. An atherogenic diet decreases liver fxr gene expression and causes severe hepatic steatosis and hepatic cholesterol accumulation: Effect of endurance training. Eur. J. Nutr. 2013, 52, 1523–1532. Chusyd D.E., Wang D., Huffman D.M., Nagy T.R. Relationships between rodent white adipose fat pads and human white adipose fat depots. Front. Nutr. 2016, 3, 10. 

Point 2: Considering the HFD model induce fatty liver also depends on insulin resistance, the authors seem to have missed examining the insulin resistance status during the treatment.

Response 2: Thank you very much for your considerable comment. Actually, we examined the insulin resistance status and oral glucose tolerance test. The results showed the potential of An Oli in improving diabetes. We would like to add the results into the supplementary data.

Figure 1. An-oligomer treatment improved metabolic parameters of diabetes and obesity in high-fat diet fed mice. (A) Plasma glucose (mmol/l) following an oral glucose load (1 g/kg) in control diet (con), high fat diet (HFD), high fat diet and PA fed (HFD+PA), high fat diet and Orlistat fed (HFD+Orlistat) mice for 4 weeks. (B) represents the area under curve (AUC) of the same groups. (C) Plasma insulin concentration (pmol/l) 30 minutes before (-30) and 15 minutes after (15) oral glucose administration of the same groups. (D) Glucose induced-insulin secretion after oral glucose administration. (E) Insulin resistance index. (F) The homeostasis model assessment of insulin resistance (HOMA-IR). (G) The quantitative insulin sensitivity check index (QUICKI). Data are mean±SEM. #p < 0.05, ## p < 0.01 compared with the Con group. *p < 0.05, ** p < 0.01 compared with the HFD group. (n = 5 per group).

Point 3: The authors need to clarify at least in the discussion how the potential mechanism of grape extract mediated down regulation in different genes. Performing the experiments mentioned in point 1 and 2 will help understand the exact role. Based on the outcome the authors need to modify the discussion.

Response 3: Thank you very much for your considerable comment. As seen in Figure 5, An-Oli helps prevent “Two hits ” pathways in NAFLD through a variety of mechanisms , such as : (1) 1st hit: depressing lipid accumulation through down-regulating lipogenesis factors (the adiponectin levels in the An-Oli fed mice were significantly higher. An-Oli reduced mRNA and protein levels of multiple lipid accumulation associated transcripts. An-Oli treatment led to reduction of DGAT, ACS, FAS, HMGCR, PPARγ and C/ EBPα mRNA levels whereas AMPK, SREBP-1c, FAS, ACC, PPARγ, FABP4 and C/ EBPα protein level were enhanced in the NAFLD mice); (2) 2nd hit: ① inhibiting activation of inflammatory pathways (mRNA expression of TNF-α, IL-1β, IL-6, IL-10, and MCP-1 were significantly decreased in An-Oli and Orlistat treated groups as compared to that in the NAFLD group. SIRT1 expression was significantly increased in An-Oli and Orlistat groups as compared to the NAFLD group. Besides, the concentration of leptin in the NAFLD with An-Oli groups displayed a statistically significant reduction compared to the NAFLD group), ② depressing oxidative stress through increased antioxidant levels (mRNA expression of Nrf2, SOD, CAT, GPx and GR expression was significantly increased in the An-Oli and Orlistat groups as compared to the NAFLD group), ③ increasing β-oxidation to inhibit mitochondrial dysfunction (mRNA expression of UCP3, CPT-1, and PGC-1α expression was significantly increased in the An-Oli and Orlistat groups as compared to the NAFLD group). We have elaborated these potential mechanisms in chapter 4.3-4.6. We would appreciate it if you would take the time to read it, and hope you satisfy our explanation.

Point 4: Along with liver weight I would like to see liver weight to body weight ratios. Preferably in the same figure.

Response 4: Thank you very much for your considerable comment. We added the liver weight to body weight ratios in figure 2 as your recommendation.

Point 5: In figure 2 along with HE an oil red O staining will help strengthen the claims. Considering the effect on adiponectin.

Response 5: Thank you very much for your considerable comment. We agree with you. Unfortunately, for some reasons, we only made paraffin sections in histopathology. Regarding adiponectin, we focused on its effect on lipid metabolism. Adiponectin concentration has been found to be correlated with lipoprotein metabolism; especially, it is associated with the metabolism of high-density lipoprotein (HDL) and triglyceride (TG). Adiponectin appears to increase HDL and decrease TG [1]. Adiponectin increase the utilization of glucose and fatty acids in the liver and skeletal muscle via adenosine monophosphate-activated protein kinase (AMPK) activation [2]. In this paper, while measuring adiponectin, we also investigated total cholesterol (TC), low-density lipoprotein cholesterol (LDL), HDL, TG and AMPK. Experimental results indicated that the levels of serum TG, TC, LDL, AMPK in the An-Oli group were significantly reduced compared with the NAFLD group. On the contrary, HDL levels were increased. These results could support the effect of adiponectin in regulating lipid metabolism. We are planning to do oil red O staining in the next investigation as your comment. Hope you understand our situation.

Yanai, H., Yoshida, H. Beneficial effects of adiponectin on glucose and lipid metabolism and atherosclerotic progression: Mechanisms and perspectives. International journal of molecular sciences, 2019, 20, 1190. Matsuda, M.; Shimomura, I. Roles of adiponectin and oxidative stress in obesity-associated metabolic and cardiovascular diseases. Rev. Endocr. Metab. Disord. 201415, 1–10.

Point 6: Authors should include genes involved in retinoic acid signalling pathway.

Response 6: Thank you very much for your considerable comment. We agree with you. Retinoic acid (RA) is a potent inhibitor on adipocyte differentiation via activation of the nuclear receptor PPARγ, a heterodimer with retinoid X receptor (RXR) [1,2]. In this paper, we investigated PPARγ signalling pathway. We would appreciate it if you would take the time to read it, and hope you satisfy our explanation.

Saeed, A., Dullaart, R., Schreuder, T., Blokzijl, H., Faber, K. Disturbed vitamin A metabolism in non-alcoholic fatty liver disease (NAFLD). Nutrients, 2018. 10, 29. de Almeida, N. R., Conda‐Sheridan, M. A review of the molecular design and biological activities of RXR agonists. Medicinal research reviews. 2019.

Point 7: Along with inflammatory gene a F4/80 IHC would be great! this will help us understand the source of inflammation and exact effect.

Response 7: Thank you very much for your considerable comment. We absolutely agree with you. However, at this time, we should revise the manuscript just in 10 days. We are planning to examine the gene in a follow-up experiment as your comment. Hope you understand our situation.

Point 8: Presentation of Figure 4 could be improved by grouping all inflammatory gene in one panel and oxidative in same panel.

Response 8: Thank you very much for your considerable comment. We revised the Figure 4 as your comment. Once again, appreciate your valuable comment.

Reviewer 3 Report

This manuscript by Meiqi Fan investigated the therapeutic potential of anthocyanin oligomers on non-alcoholic fatty liver disease model in mice. Especially estimated the Total cholesterol (TC)and triglyceride (TG) levels and the pathological changes in the liver, white adipose tissue of mice. Compared to the control group and NAFLD group, An-Oli effectively reduced TC and LDL-C, improve the liver function of NAFLD mice, regulating the blood lipids, reducing liver fat accumulation, and regulating lipid metabolism.

The An-Oli was used in study is a mixture of dimers, trimers, tetramers, and pentamer of oligomers. What method the author used to recognize all of them?

why do author use male mice, but not half male and half female mice?Is gender effect the result?

line 122 “tissue blocks were cut to a thickness of 4mm and stained with hematoxylin and eosin (H&E).” thickness of 4mm is too thick for tissue sections?

Why not compare anthocyanin monomer with An-Oli? Is the anthocyanin monomer has the same efficacy and mechanism?

Author Response

This manuscript by Meiqi Fan investigated the therapeutic potential of anthocyanin oligomers on non-alcoholic fatty liver disease model in mice. Especially estimated the Total cholesterol (TC) and triglyceride (TG) levels and the pathological changes in the liver, white adipose tissue of mice. Compared to the control group and NAFLD group, An-Oli effectively reduced TC and LDL-C, improve the liver function of NAFLD mice, regulating the blood lipids, reducing liver fat accumulation, and regulating lipid metabolism.

Thank you for your valuable time to review our manuscript. We are very much honored it and learned a lot from your supportive comments and useful suggestions. We also thoroughly revised whole manuscript as your comments, and our point-by-point reply to your comments was described as follows. In addition, other minor errors that we found while revising the manuscript were also corrected. Regarding the name of anthocyanin, according to our repeated thinking, we decided that, compared to the ‘Anthocyanin Oligomer’, ‘Polymerized Anthocyanin’ was more appropriate for the sample name (Please see Fig. 1). We have revised it in this paper and hope to get your understanding. For your convenience, we will use the name An-Oli in this response.

Point 1: The An-Oli was used in study is a mixture of dimers, trimers, tetramers, and pentamer of oligomers. What method the author used to recognize all of them?

Response 1: Thank you very much for your considerable comment. We revised the description of An-Oli preparation. After your comment, we hardly tried to explain accurately that part. The average molecular weights and its distribution of anthocyanin monomer (An-mono) and An-Oli were measured by gel permeation chromatography (GPC, Tosoh, Germany) using an sodium nitrate (0.02 N, pH 7) as elution solvent. The samples were prepared following method; 3 mg each An-mono and An-Oli was dissolved in 1 mL of sodium nitrate, and they were then filtered by 0.2 μm syringe filter. The 10 μL of final sample was injected and measured to under 40ºC and flow rate of 0.35 mL/min condition. As shown in Figure 1B and C, the molecular weight of the An-mono was found to be 788 Da, whereas the molecular weight of the An-Oli was 2,255 Da. In addition, polydispersity (PDI) which show a homogeneous molecular weight distribution was 1.28. That means the An-Oli contained homogeneous molecular weight distribution, and the anthocyanin was consistently polymerized.

Point 2: Why do author use male mice, but not half male and half female mice?Is gender effect the result?

Response 2: Thank you very much for your considerable comment. Sex and age differences between male and female C57B/Bl6 mice on HFD feeding have been reported. Male mice fed HFD on average consume more energy and therefore weigh more than their chow-fed male controls as well as female mice fed HFD [1,2]. In the longitudinal study by Ito et al [3], chronic administration of an HF diet (60% of calories from fat) caused steatohepatitis in male C57BL/6J mice. After referring to these data, we selected four-week-old male C57bl6/J mice for the experiment.

Reid, D. T., & Eksteen, B. (2015). Murine models provide insight to the development of non-alcoholic fatty liver disease. Nutrition research reviews, 28(2), 133-142. Yang, Y., Smith Jr, D. L., Keating, K. D., Allison, D. B., & Nagy, T. R. (2014). Variations in body weight, food intake and body composition after long‐term high‐fat diet feeding in C57BL/6J mice. Obesity, 22(10), 2147-2155. Ito, M., Suzuki, J., Tsujioka, S., Sasaki, M., Gomori, A., Shirakura, T., ... & Kanatani, A. (2007). Longitudinal analysis of murine steatohepatitis model induced by chronic exposure to high‐fat diet. Hepatology Research, 37(1), 50-57.

Point 3: line 122 “tissue blocks were cut to a thickness of 4mm and stained with hematoxylin and eosin (H&E).” thickness of 4mm is too thick for tissue sections?

Response 3: Thank you very much for your considerable comment. We are very sorry. That is our mistake. The experiments were conducted with 4 µm, and we corrected it after your comment.

Point 4: Why not compare anthocyanin monomer with An-Oli? Is the anthocyanin monomer has the same efficacy and mechanism?

Response 4: Thank you very much for your considerable comment. We agree with you. Anthocyanin contained in most food materials is in a monomer form but is unstable at neutral and alkaline pH and is also weakly resistant to light and heat. The polymer form is present in small amounts in foods but has higher functionality and stability than the monomer, and the typical antioxidant function thereof is also doubled [1]. For these reasons, we gave up the experiment of anthocyanin monomer for our convenience. However, after your comment, we are planning to compare the efficacy and mechanism of them for the accurate investigation. Thank you very much for your valuable comment.

Park, P. J., Jeong, T. R., Yang, H. P., & Hwang, J. W. (2018). U.S. Patent Application No. 16/046,012

Round 2

Reviewer 1 Report

Hello,

Thank you for accepting comments and making changes to your manuscript.  Materials and Methods are now clear to me, and the data is also clearly presented.  I have no additional comments.

Sincerely,

Author Response

Response to Reviewer 1Comments

Hello,

Thank you for accepting comments and making changes to your manuscript.  Materials and Methods are now clear to me, and the data is also clearly presented. I have no additional comments.

Sincerely,

→ Thank you for your valuable time to review our manuscript. We are very much honored it and learned a lot from your supportive comments and useful suggestions. Once again, appreciate your time.

Reviewer 2 Report

The authors have answered most of my doubts. However, I am still not satisfied with the discussion. The authors need to discuss what was first triggered. Since the inflammatory reduction of oxidative damage may be due to reduction in the first hit not in the second hit. Hence  I am still unsure what is the FIRST TRIGGER OF PROTECTIVE RESPONSE. 

The discussion provided here is just a summary of finding and extrapolation. The readers will be interested in why this extract worked so good and at what stage.

Author Response

Response to Reviewer 2Comments

The authors have answered most of my doubts. However, I am still not satisfied with the discussion.

→ Thank you for your valuable time to review our manuscript. We are very much honored it and learned a lot from your supportive comments and useful suggestions. We also thoroughly revised whole manuscript as your comments, and our point-by-point reply to your comments was described as follows. In addition, other minor errors that we found while revising the manuscript were also corrected.

Point 1:The authors need to discuss what was first triggered. Since the inflammatory reduction of oxidative damage may be due to reduction in the first hit not in the second hit. Hence I am still unsure what is the FIRST TRIGGER OF PROTECTIVE RESPONSE.

Response 1: Thank you very much for your considerable comment.We thought PPARγ was the first trigger, and described it in the discussion section and Figure 5. However, after your comment, we described in detail about that to be clear. Thanks to you, our investigation would be upgraded. Hopefully, you satisfy our revision.

Point2:The discussion provided here is just a summary of finding and extrapolation. The readers will be interested in why this extract worked so good and at what stage.

Response 2: Thank you very much for your considerable comment. We revised the discussion section as your comment. Once again, appreciate your valuable time.
